# Utilizing the SEIPS model to guide hand hygiene interventions at a tertiary hospital in Ethiopia

Leigh Berman[1]*, Meredith Kavalier[2], Beshea Gelana[3], Getnet Tesfaw[4], Dawd Siraj[2], Daniel Shirley[2], Daniel Yilma[5,6]

**1** University of Wisconsin School of Medicine and Public Health, Madison, WI, United States of America, **2** Division of Infectious Disease, Department of Medicine, University of Wisconsin School of Medicine and Public Health, Madison, WI, United States of America, **3** Department of Health Policy and Management, Jimma University, Jimma, Ethiopia, **4** School of Medical Laboratory Sciences, Jimma University, Jimma, Ethiopia, **5** Department of Internal Medicine, Jimma University, Jimma, Ethiopia, **6** Jimma University Clinical Trial Unit, Jimma University, Jimma, Ethiopia

☯ These authors contributed equally to this work.
* lrberman@wisc.edu

**Data Availability Statement:** All relevant data are within the manuscript and its Supporting Information files.

## Abstract

We aimed to apply the Systems Engineering Initiative for Patient Safety (SEIPS) model to increase effectiveness and sustainability of the World Health Organization's (WHOs) hand hygiene (HH) guidelines within healthcare systems. Our cross-sectional, mixed-methods study took place at Jimma University Medical Center (JUMC), a tertiary care hospital in Jimma, Ethiopia, between November 2018 and August 2020 and consisted of three phases: baseline assessment, intervention, and follow-up assessment. We conducted questionnaires addressing HH knowledge and attitudes, interviews to identify HH barriers and facilitators within the SEIPS framework, and observations at the WHO's 5 moments of HH amongst healthcare workers (HCWs) at JUMC. We then implemented HH interventions based on WHO guidelines and results from our baseline assessment. Follow-up HH observations were conducted months later during the Covid-19 pandemic. 250 HCWs completed questionnaires with an average knowledge score of 61.4% and attitude scores indicating agreement that HH promotes patient safety. Interview participants cited multiple barriers to HH including shortages and location of HH materials, inadequate training, minimal Infection Prevention Control team presence, and high workload. We found an overall baseline HH compliance rate of 9.4% and a follow-up compliance rate of 72.1%. Drastically higher follow-up compared to baseline compliance rates were likely impacted by our HH interventions and Covid-19. HCWs showed motivation for patient safety despite low HH knowledge. Utilizing the SEIPS model helped identify institution-specific barriers that informed targeted interventions beyond WHO guidelines aimed at increasing effectiveness and sustainability of HH efforts.

**Funding:** This project is supported by Jimma University Medical Center (DY), the Jimma University Clinical Trial Unit (DY), and the University of Wisconsin School of Medicine and Public Health Shapiro Summer Research Program (LB, summerresearch.med.wisc.edu). This project was also partially funded through a Faculty and Staff Travel Award from the University of Wisconsin-Madison Global Health Institute (DSh). The funders had no role in study design, data collection and analysis, decision to publish, or preparation of the manuscript.

**Competing interests:** The authors have declared that no competing interests exist.

## Introduction

Health care associated infections (HAIs) are common but avoidable adverse patient events that pose great threats to patient safety and increase morbidity, mortality, and length of hospital stays [1]. Although HAIs affect millions of patients globally, a 2011 meta-analysis of international literature showed that the average prevalence of HAIs in low to middle-income countries (LMICs) is 3 times higher than in high-income countries [2].

Hand hygiene (HH) is an inexpensive and effective method for reducing HAIs, disease outbreaks, and antimicrobial resistance [2–4]. Despite this knowledge, reports from LMICs and Ethiopia specifically consistently show HH compliance rates under 25% [2, 5, 6]. The World Health Organization (WHO) has recognized the global need for improved HH and published guidelines for implementation HH interventions for LMICs [7]. While there is abundant global evidence that this valuable, although generic tool can help improve HH compliance rates—including at two teaching hospitals in Ethiopia—significant barriers to and after implementation of WHO HH interventions exist in LMICs [8–13]. Further, without additional efforts aimed at sustainability of WHO interventions, studies show HH compliance may decrease over time after initial improvement [14]. A 2015 meta-analysis and a 2017 systematic review demonstrated that implementation of WHO guidelines alongside another intervention strategy that takes human behavioral theory into account, such as incentives, goal setting, and accountability results in greater improvement in HH compliance rates [15, 16].

The Systems Engineering Initiative for Patient Safety (SEIPS) model has been widely used around the world to guide quality improvement projects by illustrating how five components of the work system—person, tools and technology, organization, environment, and task—interact to impact patient safety measures such as HH [17–19]. We aimed to apply the SEIPS model to increase effectiveness and sustainability of HH interventions at Jimma University Medical Center (JUMC) in Jimma, Ethiopia. To our knowledge, no other studies have used the SEIPS model as a supplemental strategy to improve implementation of WHO HH recommendations.

This project is part of a larger collaboration between JUMC and the University of Wisconsin School of Medicine and Public Health aimed at establishing and improving infection control practices [19].

## Materials and methods

This study was conducted at JUMC in Jimma, Ethiopia. JUMC is the only teaching and referral hospital in the southwestern part of the country and serves a catchment population of over 15 million. The JUMC workforce includes 1053 HCWs: 218 physicians, 603 nurses, 82 midwives, 78 laboratory technicians, and 72 pharmacy professionals. JUMC has an Infection Prevention and Control (IPC) team established in 2017. Our study was conducted between November 2018 and August 2020.

We conducted an institution-based, cross-sectional study consisting of three phases: baseline assessment, interventions, and follow-up assessment.

### Phase I: Baseline assessment

From November to December 2018, our collaborative team developed a study protocol and a questionnaire to assess baseline HH knowledge, attitudes, and hospital infrastructure. The JUMC IPC team distributed these questionnaires from December 2018 to February 2019. In June 2019, the IPC team conducted observations of HH practices throughout the hospital. In July 2019, we conducted qualitative interviews amongst HCWs regarding barriers and

facilitators to HH. All questionnaires and forms were written in English, the official language of medical education in Ethiopia.

**Knowledge and attitudes questionnaire.** Our questionnaire consisted of three sections: participant demographics, HH knowledge, and attitudes and perceptions about HH (S1 Appendix). Knowledge questions included 25 questions from the WHO Hand Hygiene Knowledge Questionnaire for Healthcare Workers and two additional knowledge questions adapted from the Institute for Healthcare Improvement's "A Guide for Improving Practices among Health Care Workers" [20]. The attitude and perception questions used a Likert 5-scale and were adapted from the WHO HH Perception Survey for Healthcare Workers to fit the local context. Questionnaires were distributed to HCWs on 8 different primary wards over the span of two months by systematic sampling within professional categories. Eligibility criteria included (1) HCWs at JUMC (2) able to read and write in English, and participants were recruited by word-of-mouth. Sample size was determined using a formula (Eq 1) for estimating single population proportions under the following assumptions. To obtain the largest sample size, we used a p value of 0.5 (p) = 50%, a Z value (Z a/2) with a 95% confidence level (1.96), and a margin of error of 0.05 (d):

$$n = \frac{\left(\frac{Za}{2}\right)^2 p(1-p)}{d^2} = \frac{(1.96)^2 0.5(1-0.5)}{0.05^2} = 384 \tag{1}$$

Since our source population was less than 10,000, we then used a finite population correction formula based on the number of health care professionals at JUMC, 1053. Adding 10% for non-response rate gave a final target sample size of 309.

Completed questionnaires were collected by study staff and recorded in Excel. Each participant's composite knowledge score was calculated out of 27 possible points. For attitude questions, responses were entered according to the Likert 5-scale spectrum.

**Baseline observations.** Observations of HH practices were conducted by IPC observers who completed training in HH. Observations took place on workdays (Monday-Friday) over 2-months. HCWs were selected for observation via systematic sampling within professional categories. Observers used the WHO's HH observation form to capture HH opportunities for and compliance with HH (washing hands with soap and water or using alcohol hand rub) at each of the WHO's 5 moments of hand hygiene: before touching a patient, before clean/aseptic procedures, after body fluid exposure, after touching a patient, and after touching patient surroundings. While conducting observations, observers did not identify themselves or their role to reduce HCWs' awareness that they were being observed.

**Qualitative interviews.** We conducted semi-structured interviews amongst HCWs at JUMC able to converse in either English or Amharic. Participants were recruited by word-of-mouth using convenience sampling. Literature review revealed a minimum of 12 interviews would attain theoretical saturation [21]. Each participant was given a written information sheet describing our study and asked about their demographic information prior to the interview. We utilized an interview guide containing both open and close ended questions developed within the SEIPS framework by study staff to assess perceived barriers and facilitators to HH (S2 Appendix) [17]. Interviews were conducted primarily by two study staff members (LB and GT) in English with translation in Amharic if needed and took place in private workspaces in JUMC. Interviews lasted between 20–40 minutes depending on participant responses. The interviews were recorded, manually transcribed, and thematically coded using QRS Nvivo (Version 12.4.0) in accordance with the SEIPS model by one researcher. Quotations were coded as "barrier," "facilitator," "attitude," and/or "current practice" and as one or multiple of the 5 SEIPS components of the work system.

## Phase II: Intervention

Between August and September 2019, after completion of the baseline assessment, the IPC team enacted educational, resource, and environment interventions in accordance with WHO guidelines and based on initial findings from interviews. The IPC team installed 400 hand rub dispensers on the walls inside of multiple occupancy patient rooms and in ward hallways, placed posters promoting HH around the hospital, and added 10 TVs providing educational messages about HH in main hospital corridors. One IPC coordinator, two full-time IPC officers, and thirty-two staff members from various units of the hospital selected as IPC Focal Persons were tasked with monitoring IPC activities and providing IPC training in their units. IPC staff underwent five days of training on the basics of infection prevention including HH in preparation for their new roles. Then, 184 nurses underwent day long HH training sessions conducted by IPC team leaders.

## Phase III: Follow-up

HH observations were repeated between May and August 2020 using the same methods previously described for the baseline observations.

## Data analysis

All categorical data were summarized as proportions and frequencies and compared using chi-squared tests or Fisher's exact tests. We reported continuous data as mean plus standard deviation and compared them using one-way ANOVA. All Graphical analyses were conducted using slide plots. P-values less than or equal to 0.05 were considered statistically significant. We used STATA version 16 SE for data analyses. For outcome data, we calculated an odds ratio and 95% confidence interval to compare HH compliance in the follow-up and baseline assessment.

## Ethical considerations

Ethical approval was obtained from both Jimma University and the University of Wisconsin Madison Institutional Review Boards. Prior to interview and questionnaire completion, an information sheet was provided to each participant and verbal consent was recorded by study staff. The ethics committee at both JUMC and the University of Wisconsin approved the verbal consent procedure and agreed written consent was not needed.

# Results

## Questionnaires

Two hundred fifty-one baseline questionnaires were analyzed, with one participant excluded due to survey incompleteness (Table 1) [22]. The following professional categories at JUMC were represented: nurse/midwife (62%), physicians (28.8%), and laboratory technicians who function as phlebotomists (9.2%). Participants worked on 8 different wards or units including medicine (24.8%), surgery (25.2%), pediatrics (14.8%), Gyn-Ob (21.6%), laboratory (9.2%), ICU (2.4%), outpatient department (1.6%), and emergency department (0.4%). Across all groups, 13.6% of participants reported prior HH training, despite no systematic training at JUMC prior to this study, and 69.9% reported routine compliance with HH. While there was no statistical difference in perceived HH compliance across professional categories (p = 0.11), trained participants report significantly higher HH compliance (88.2%) than untrained participants (66.7%) (p = 0.009).

**Knowledge.** The average baseline HH knowledge score amongst all participants was 16.6 ± 2.9 out of 27, indicating participants answered an average of 61.4% of questions correctly. On average, physicians scored highest (17.6 ± 3.2), followed by laboratory technicians

**Table 1. Questionnaire participant demographics, knowledge, and attitudes between professional categories.**

|  | Total (N = 250) | Nurse/ Midwife (N = 155) | Physician (N = 72) | Laboratory Tech (N = 23) | P value |
|---|---|---|---|---|---|
| Age, mean (SD) | 27.2 (5.1) | 25.8 (4.5) | 29.5 (5.6) | 29.1 (4.7) | **<0.001**[*a] |
| Sex, n (%) |  |  |  |  |  |
| Male | 149 (59.6) | 73 (47.1) | 59 (81.9) | 17 (73.9) | **<0.001**[*b] |
| Female | 101 (40.4) | 82 (52.9) | 13 (18.1) | 6 (26.1) |  |
| Received HH training in last 3 years, n (%) |  |  |  |  |  |
| Trained | 34 (13.6) | 26 (16.8) | 8 (11.1) | 0 (0) | 0.06[c] |
| Untrained | 216 (86.4) | 129 (83.2) | 64 (88.9) | 23 (100) |  |
| HH practices and knowledge |  |  |  |  |  |
| Reports performs HH routinely, n (%) | 174 (69.6) | 108 (69.7) | 46 (64.9) | 20 (87) | 0.11[b] |
| Knowledge Total, mean (SD) | 16.6 (2.9) | 16.2 (2.6) | 17.6 (3.2) | 16.3 (2.7) | **0.002**[*a] |

a = ANOVA,

b = chi-squared,

c = Fisher's exact test.

*Statistically significant at p<0.05.

(16.3 ± 2.7), and then nurses/midwives (16.2 ± 2.6) as shown in Table 1 and Fig 1. There was no statistically significant difference in knowledge scores between participants that identified as trained (16.2 ± 2.5) vs untrained (16.6 ± 2.9) (Fig 2).

**Attitudes and perceptions.** Table 2 outlines differences in HH attitudes and perceptions between professional categories. Physicians perceived HH to be more effective in reducing patient mortality and healthcare costs than other groups. Nurses/midwives felt less strongly than other groups that HH is a habit of their personal lives and that the demands of their job require prioritization of other tasks over HH.

## Interviews

Of the 22 interview participants, 59.1% were medical practitioners (resident physicians and medical interns), 36.4% were nurses, and 4.5% were midwives. Participants worked in a variety of wards including internal medicine (40.9%), pediatrics (22.7%), neonatal ICU (9.1%), and OB/Gyn (9.1%). Over two-thirds of participants stated they rarely perform HH, and 68% of interviewees stated they are more likely to perform HH if a patient appears infectious. Only 21% of participants had taken part in prior HH training, and only 38% knew JUMC had an IPC team. Participants cited multiple barriers and facilitators to HH within the SEIPS components of the work system (Tables 3 and 4).

**Tools and technology.** All participants cited shortages of water, functioning sinks, soap, and alcohol hand rub. Although all interview participants worked in wards with some dispensers installed, the vast majority of participants stated that the number of dispensers was inadequate, and nearly one fourth of participants said they had not seen any dispensers at all. All interviewees who had seen the dispensers stated that they were rarely filled with hand rub. Nevertheless, facilitators within this category included recent increased availability of alcohol hand rub due to placement of some dispensers (45%) and utilization of independently purchased pocket hand rub mostly by resident physicians (55%).

**Person.** Participants identified inadequate HH training (95%) and lack of awareness (41%) as barriers to HH. Knowledge of the purpose and importance of HH in preventing HAIs (86%) and informal HH education by superiors (36%) particularly by attending physicians during rounds were discussed as facilitators.

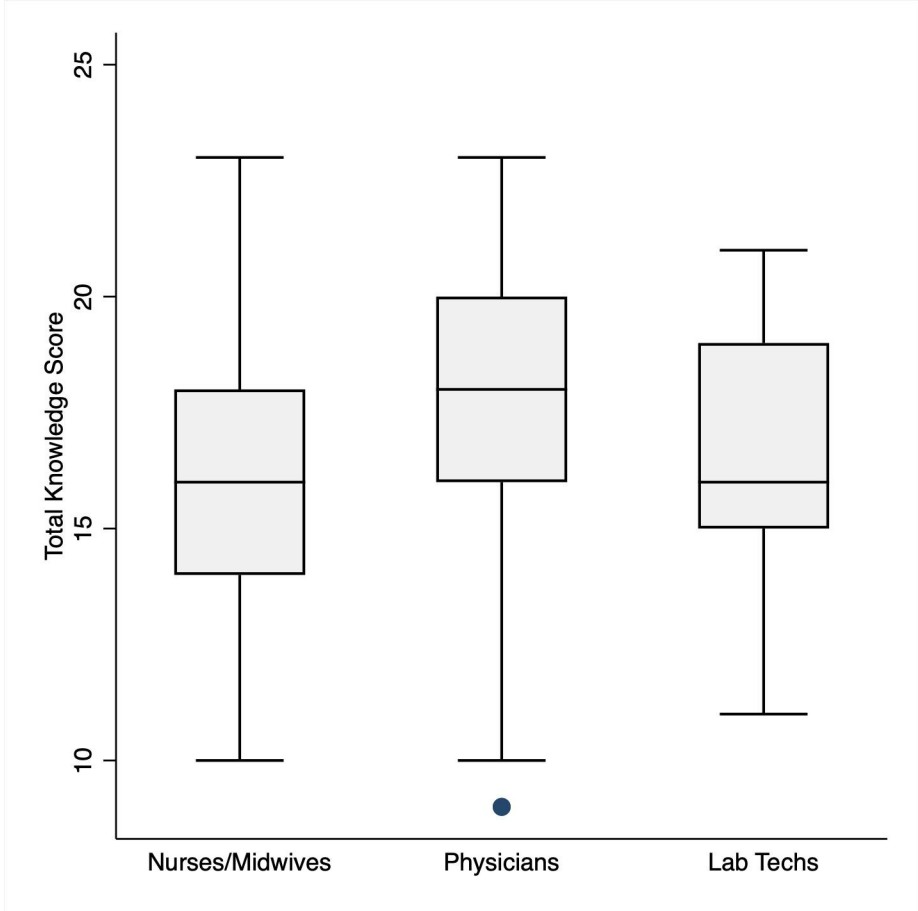

**Fig 1. Knowledge scores by professional category.** Mean knowledge scores amongst nurses/midwives (16.2 ± 2.6), physicians (17.6 ± 3.2), and laboratory technicians (16.3 ± 2.7) at Jimma University Medical Center (p-value 0.002).

**Organization.** Organizational barriers included lack of HH monitoring and surveillance (100%), perceived lack of IPC or organized body focusing on HH initiatives (32%), and inadequate dispensing and management of supplies by both the national dispensary and JUMC (27%). 18% of interviewees discussed hospital norms as a barrier to HH. Acting as a facilitator, 68% of participants stated that they would feel comfortable advising a colleague to perform HH if they noticed poor HH compliance, illustrating a positive patient safety culture at JUMC.

**Task.** Barriers included high workload and high patient volume (64%). No "task" facilitators were identified.

**Environment.** 36% of participants mentioned HH posters as facilitators, and 82% indicated the need for HH posters throughout the hospital. Another barrier within this category was the location of HH materials (55%). Participants stated that alcohol hand rub and gloves are frequently stored at the nursing station rather than patient rooms, making access to HH materials difficult, particularly at night when nursing stations are locked.

## Observations

We observed 1386 opportunities for HH in the baseline assessment and 639 in the follow-up assessment (Fig 3) [22]. Our follow-up total HH compliance of 72.1% was significantly higher than our baseline total HH compliance of 9.4% (odds ratio 25.0; 95% confidence interval 19.5 to

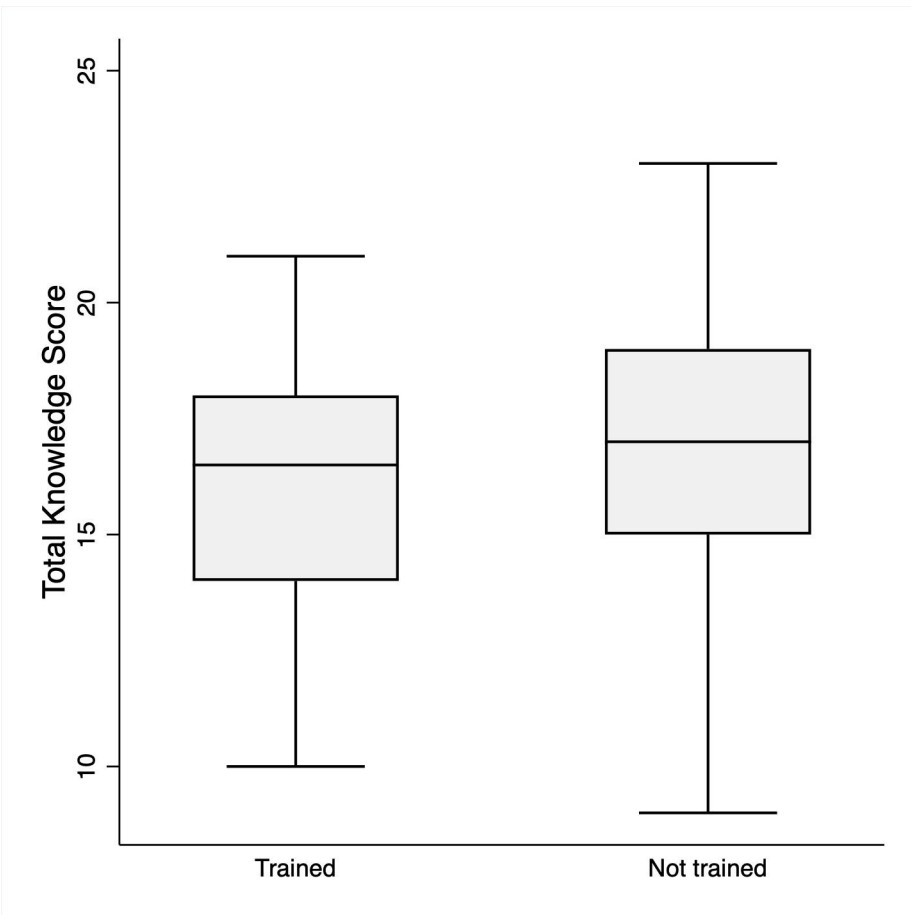

**Fig 2. Knowledge scores in trained vs untrained healthcare workers.** Mean knowledge scores for trained (16.2 ± 2.5) vs untrained (16.6 ± 2.9) participants at Jimma University Medical Center (p-value 0.41).

32.1). Physicians had the highest baseline compliance (12.4%), followed by nurses (8.7%), and then midwives (6.4%). Midwives had the highest follow-up compliance (75.0%), followed by nurses (74.4%), and then physicians (65.0%). Baseline HH compliance was higher after contact with patient surroundings (27.9%) compared to the other moments of HH. Follow-up HH compliance was higher after contact with patients (79.2%), their surroundings (77.7%), and bodily fluids (87.4%) compared to before patient contact (53.8%) and aseptic procedures (63.8%).

## Discussion

Our study found significantly higher HH compliance amongst all professional categories during the follow-up period compared to the baseline period. Notably, our follow-up HH observations took place during the Covid-19 pandemic. The pandemic likely contributed to the drastically improved HH compliance at JUMC for a number of reasons including increased availability of HH materials from the national dispensary, enhanced IPC training on Covid-19, increased awareness of the importance of infection prevention, and decreased workload due to lower admission rates. Other studies around the world have similarly shown increased HCW HH during the pandemic [23]. Although the timing of follow-up observations inhibits our ability to fully attribute increased compliance rates to IPC educational and resource interventions, follow-up compliance rates demonstrate that when HH materials are available and

**Table 2. Attitude scores by professional category.**

| Attitude Statement | Nurse/ Midwife (N = 155) | Physician (N = 72) | Laboratory Tech (N = 23) | P value |
|---|---|---|---|---|
| *I am tasked to act as a model about hand hygiene for other healthcare personnel* | 3.14 (1.28) | 3.61 (1.39) | 2.87 (1.42) | **0.02*** |
| *Execution of hand hygiene may reduce mortality of patients under the recommended conditions* | 3.4 (1.12) | 4.15 (1.22) | 3.69 (1.14) | **<0.001*** |
| *Execution of hand hygiene may reduce the related medical costs to nosocomial infections under the recommended conditions* | 3.46 91.28) | 4.2 (1.26) | 3.78 (1.17) | **<0.001*** |
| *Prevention from the acquired infections is deemed as one of valuable roles for personnel of healthcare services.* | 3.49 (1.19) | 4.2 (1.25) | 3.9 (1.22) | **<0.001*** |
| *The existing infectious diseases in healthcare-giving environments may threaten my life and occupation.* | 2.5 (1.16) | 1.74 (1.16) | 2.26 (1.32) | **<0.001*** |
| *I think I have the potential to change poor performances regarding hand hygiene in my workplace* | 3.55 (1.22) | 3.99 (1.33) | 3.74 (1.54) | 0.06 |
| *The hand hygiene is assumed as a habit in my personal life.* | 3.42 (1.37) | 3.99 (1.22) | 4.0 (1.43) | **0.005*** |
| *It is more important for me to fulfill perfectly my tasks than doing hand hygiene when the ward is busy* | 3.1 (1.18) | 3.63 (1.38) | 3.6 (1.5) | **0.007*** |
| *I could not always do hand hygiene under the recommended situations because of preference of my patients' requirements.* | 3.22 (1.27) | 3.83 (1.24) | 3.78 (1.47) | **0.002*** |
| *I think one could follow the medical service officials in order to make decision for execution and or non- execution of hand hygiene* | 3.18 (1.26) | 2.85 (1.52) | 2.78 (1.44) | 0.14 |

Participants used a scale of 1–5 to indicate "1 = strongly disagree, 2 = disagree, 3 = neutral, 4 = agree, 5 = strongly agree" for each statement. Mean attitude scores reported (SD). One-way ANOVA used to calculate P value.

*Statistically significant at p<0.05.

HCWs feel able and incentivized to prevent the spread of infection, adequate HH is achievable at hospitals in LMICs like JUMC.

Baseline compliance at JUMC, while under 10%, was found to be higher than baseline compliance at two other hospitals in Ethiopia [8, 9]. This may be due to the existence of

**Table 3. Identified barriers and facilitators to HH classified within the SEIPS model's 5 components of the work system: Tools and technology, organization, person, task, and environment.**

| SEIPS Category | Barrier | N = 22 | Facilitator | N = 22 |
|---|---|---|---|---|
| **Tools and technology** | Lack of functional water facilities and/or water supply, hand rub, and soap | 22 | Pocket hand rub | 12 |
| | Lack of gloves | 21 | Dispensers increase access to hand rub | 10 |
| | | | Adequate number of sinks in wards | 10 |
| **Organization** | Lack of HH monitoring and surveillance | 22 | Communication between HCWs to promote patient safety | 15 |
| | Perceived lack of IPC or organized body focusing on HH initiatives | 7 | Head nurse manages ward supplies well | 5 |
| | Inadequate dispensing and management of supplies | 6 | | |
| | Hospital norms exclude HH | 4 | | |
| | Lack of facility maintenance | 4 | | |
| **Person** | Lack of formal training | 21 | Previous knowledge of HH | 19 |
| | Lack of awareness | 9 | Informal HH education from school instructors, peers, and superiors | 8 |
| **Task** | High workload | 14 | | |
| **Environment** | Too few HH posters | 18 | Some HH signs and posters | 8 |
| | HH materials kept outside of patient rooms | 12 | | |
| | HH materials in nursing station and bathrooms locked | 4 | | |

**Table 4. Representative quotes of themes developed from qualitative responses with the 5 SEIPS classifications.**

| Code | Theme | Representative Quote |
|---|---|---|
| | | **Tools and Technology** |
| Barrier | Lack of hand rub in wall dispensers | "Sometimes, guests come, or another foreign[er] comes for visit. For one up to two days, [dispersers are] fill[ed], then stop. That is the problem. I think today is filled because of you." (Nurse) |
| Barrier | Training is ineffective without HH materials | "Giving training without availing the material. . .when we go up to the ward, if there is no availability of the materials, [training] is the same as adding water on the stone. It is useless." (Midwifery Staff) |
| | | **Organization** |
| Barrier | Supply dispensing problems | "I don't think the problem is with the hospital. Rather, the supply chain management starting from the federal ministry of health to this hospital is difficult sometimes." (Resident Physician) |
| Barrier | Hospital norms exclude HH | "[Even having] water, you may ignore washing your hand. And even, you can see that your elders are also not doing that thing. And it seems that it is strange to wash in between your work. . .to go there and to wash your hand. So. . .even having materials to wash your hand. . .no one around is doing that, so how can I wash my hand?" (Medical Intern) |
| | | **Person** |
| Barrier | Lack of awareness | "The hospital should, I think, introduce this alcohol hand rub to us. Because I think most of us don't know, and so, the hospital should make it available and should promote us to use it so that we can use other alternatives to water." (Medical Intern) |
| Facilitator | Informal training | "When we round. . .seniors always address us about hand hygiene. They usually do this hand hygiene mechanisms. They use alcohols, other disinfectants." (Resident Physician) |
| | | **Task** |
| Barrier | High workload | "While I am working as a resident in the OPD, I may evaluate like 30 patients per day. So, crowding of the job is also one. Just washing my hands like 13 times or 14 times per day sometimes prevents me to do it frequently, just when always turning from one patient to others." (Resident Physician) |
| | | **Environment** |
| Barrier | HH posters and signs | "The posters should be posted at every site. Hand hygiene seems simple, but it can lead to difficult problems." (Nurse) |

some hand rub dispensers and HH posters around the hospital before baseline assessment. Despite a baseline HH compliance of less than 10%, the majority of questionnaire participants reported performing HH routinely. The large discrepancy between perceived and observed HH compliance likely demonstrates HCWs underestimated how frequently HH is recommended. Our finding that baseline and follow-up compliance were higher after patient contact than before patient contact is consistent with several studies from Sub-Saharan Africa and has been previously attributed to HCW's conceptualization of HH as a way to protect themselves rather than their patients from infection [6, 9, 24]. This aligns with our qualitative interview finding that HCWs are more likely to perform HH if a patient appears infectious.

While three reports from other hospitals in Ethiopia found higher or similar HH compliance rates amongst nurses than physicians, our baseline assessment revealed higher HH compliance rates amongst physicians [6, 8, 9]. Studies from other LMICs that have found higher HH compliance amongst physicians than nurses seem to attribute this to potentially stronger HH education and perceptions of professional responsibility [25]. Based on our study, multiple factors may have contributed to higher physician compliance. First, in qualitative interviews, many resident physicians but few nurses discussed purchasing their own pocket hand rub and

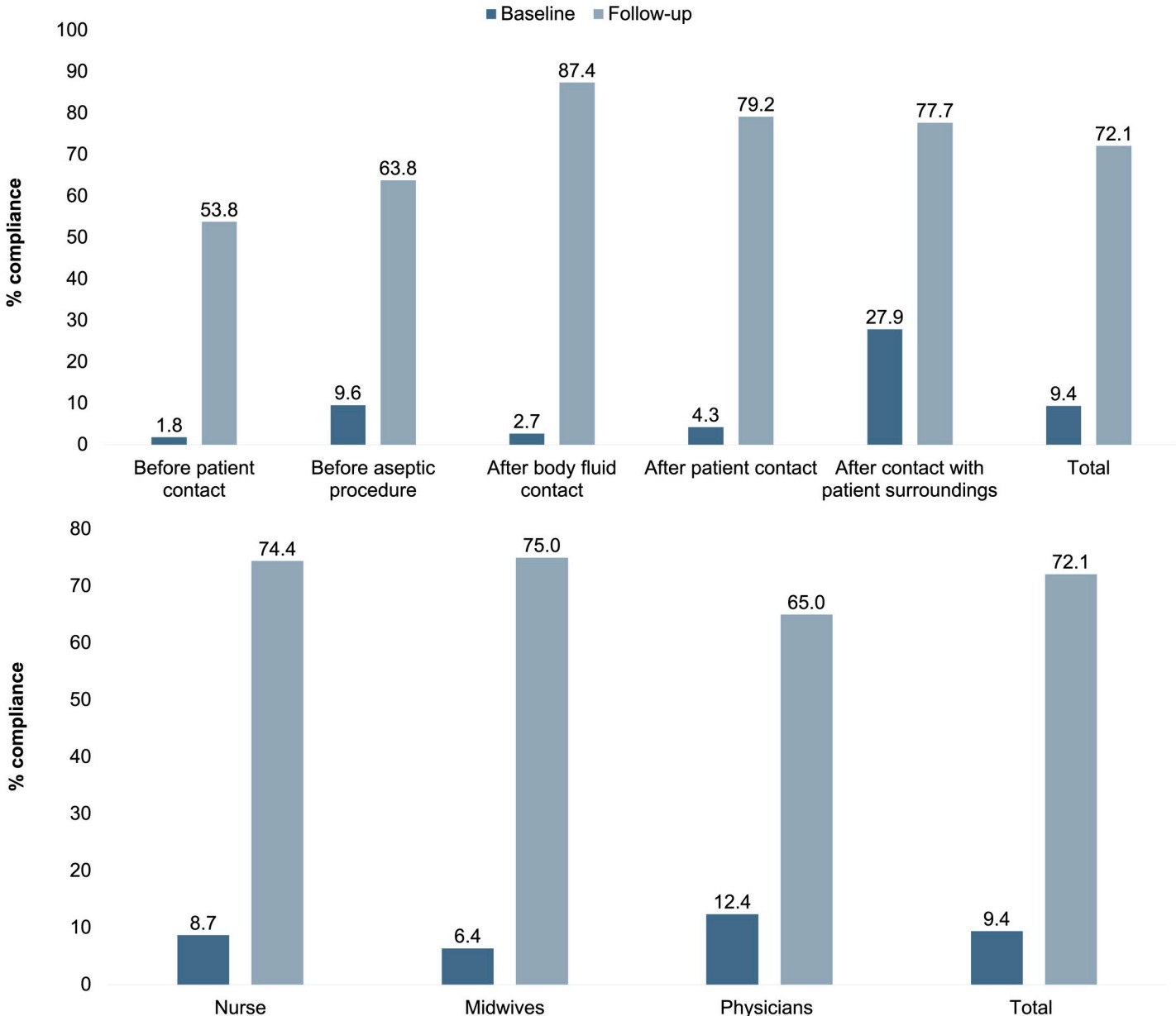

**Fig 3. Hand hygiene compliance at Jimma University Medical Center.** HH compliance (%) at the WHO's 5 moments of HH. (% compliance calculated from number of recorded HH actions and number of HH opportunities based on the WHO's 5 moments of HH).

receiving informal HH education from senior physicians on rounds. Second, attitude scores indicate greater agreement that HH prevents HAIs and a greater sense of agency to impact patient outcomes amongst physicians compared to other HCWs. Our follow-up data showing higher compliance amongst midwives and nurses than physicians may indicate that when HH resources are more equitably accessible, nurses and midwives are more conscientious about their HH than physicians.

Both the IPC's interventions and the Covid-19 pandemic likely contributed to improved HH at JUMC over the course of this study, but further interventions aimed at sustaining this progress are needed. Utilizing the SEIPS model throughout this study helped identify

actionable items beyond the WHO HH guidelines upon which the IPC team can focus to address HH via a multi-modal approach.

In line with other reports from Sub-Saharan Africa regarding barriers to HH, interviews demonstrated the need for sustainable access to alcohol hand rub at JUMC [24]. The IPC team can achieve this by placing more dispensers around the hospital and assigning and training HCWs to reliably produce hand rub when materials are available and fill empty dispensers. Consistent availability of HH materials must be available before impactful changes can occur in hospital norms and individual behavior.

Interview responses and low knowledge scores across all professional categories and regardless of prior training illustrate the importance of increased HH education for all HCWs at JUMC. Still, increased training alone is unlikely to result in sustained HH compliance without further reminders and incentives [10, 15, 16, 26]. Informal HH education from superiors may serve as a facilitator for physicians, as these reminders can occur more frequently than organized trainings and incentivize learning doctors to follow the attending physician's lead. Encouraging informal education and feedback from high-level HCWs and IPC officers may be a relatively simplistic way to provide incentives at the interpersonal level.

Increased HAI tracking and HH monitoring will aid in creating institution-based incentives for improved HH, which have been found to enhance effectiveness and sustainability of HH interventions [10, 15, 16, 26]. Reports have also shown increased HH compliance with interventions targeting social influence and leadership, emphasizing the importance of interviewee's suggestion for increased IPC team visibility within JUMC [27, 28].

Our interview results align with prior reports that have identified hospital overcrowding and shared patient spaces as a barrier to HH in LMICs [24]. Adapting WHO HH education materials used in HCW training to address the ambiguity of HH recommendations within shared "patient zones" would ensure that HH education more directly pertains to clinical practice at JUMC [29]. Also within "environment," qualitative interviews identified the need for more HH posters to serve as frequent reminders and a material storage system that ensures alcohol hand rub and gloves are available to HCWs all day. Increased availability of gloves should be accompanied by IPC training on the necessity of HH even when gloves are used, as some studies suggest glove use discourages HH [9].

Several empirical studies have shown a negative relationship between high workload and HH compliance [30, 31]. Both attitude scores and qualitative interview responses regarding "task" in our study similarly suggest the need for interventions that help HCWs perform HH despite high workload and patient volume. Further research on strategies to mitigate impact of high workload on HH are needed.

Our study has several limitations. First, this is a single-institution study, so results may not be applicable to other hospital systems. Secondly, our interview and questionnaire results are subject to inclusion bias, as all participants agreed to participate in a study about HH. Response and interviewer bias may have also influenced interview results. Thirdly, the larger sample size in baseline compared to follow-up observations makes it difficult to fairly compare compliance rates between these two periods. Lastly, the Covid-19 pandemic's impact on our study should be re-emphasized.

## Conclusion

Overall, WHO guidelines alone cannot sufficiently address all barriers to HH within a specific healthcare system. Utilizing the SEIPS model in interviews enabled us to identify additional areas of improvement and implementation gaps that are largely institution specific. Every hospital has a distinct array of factors contributing to low compliance, and the generic WHO tool

should be used in conjunction with another strategy like the SEIPS model to determine the most specific and sustainable additional interventions for a given institution. Longitudinal monitoring and evaluation of interventions will allow the IPC team to ensure sustainability of interventions and continuously identify further steps for improvement.

## Supporting information

**S1 Appendix. Hand hygiene questionnaire.**
(DOCX)

**S2 Appendix. Interview guide developed within the SEIPS framework.**
(DOCX)

## Acknowledgments

Thank you to the staff and faculty at Jimma University for supporting and participating in this project.

## Author Contributions

**Conceptualization:** Beshea Gelana, Dawd Siraj, Daniel Shirley, Daniel Yilma.

**Data curation:** Leigh Berman, Beshea Gelana, Getnet Tesfaw, Daniel Shirley, Daniel Yilma.

**Formal analysis:** Leigh Berman, Meredith Kavalier, Beshea Gelana, Daniel Yilma.

**Funding acquisition:** Dawd Siraj, Daniel Shirley, Daniel Yilma.

**Investigation:** Leigh Berman, Beshea Gelana, Getnet Tesfaw, Daniel Yilma.

**Methodology:** Beshea Gelana, Dawd Siraj, Daniel Shirley, Daniel Yilma.

**Project administration:** Beshea Gelana, Dawd Siraj, Daniel Shirley, Daniel Yilma.

**Resources:** Daniel Yilma.

**Supervision:** Beshea Gelana, Dawd Siraj, Daniel Shirley, Daniel Yilma.

**Visualization:** Leigh Berman, Meredith Kavalier.

**Writing – original draft:** Leigh Berman, Meredith Kavalier.

**Writing – review & editing:** Leigh Berman, Meredith Kavalier, Beshea Gelana, Getnet Tesfaw, Dawd Siraj, Daniel Shirley, Daniel Yilma.

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
