## [Decision Letter · Decision Letter 0]

27 Aug 2021

PONE-D-21-07299

Utilizing the SEIPS model to guide hand hygiene interventions at a tertiary hospital in Ethiopia

PLOS ONE

Dear Dr. Berman,

Thank you for submitting your manuscript to PLOS ONE. After careful consideration, we feel that it has merit but does not fully meet PLOS ONE’s publication criteria as it currently stands. Therefore, we invite you to submit a revised version of the manuscript that addresses the points raised during the review process.

We look forward to receiving your revised manuscript.

Kind regards,

Monika Pogorzelska-Maziarz

Academic Editor

PLOS ONE

1. Please ensure that your manuscript meets PLOS ONE's style requirements, including those for file naming. The PLOS ONE style templates can be found at https://journals.plos.org/plosone/s/file?id=wjVg/PLOSOne_formatting_sample_main_body.pdf and https://journals.plos.org/plosone/s/file?id=ba62/PLOSOne_formatting_sample_title_authors_affiliations.pdf.

3. Thank you for stating the following in the Acknowledgments/Funding Section of your manuscript:

“This project is supported by Jimma University Medical Center (DY), the Jimma University Clinical Trial Unit (DY), and the University of Wisconsin School of Medicine and Public Health Shapiro Summer Research Program (LB). The funders had no role in study design, data collection and analysis, decision to publish, or preparation of the manuscript.”

“This project is supported by Jimma University Medical Center (DY), the Jimma University Clinical Trial Unit (DY), and the University of Wisconsin School of Medicine and Public Health Shapiro Summer Research Program (LB, summerresearch.med.wisc.edu). The funders had no role in study design, data collection and analysis, decision to publish, or preparation of the manuscript.”

4. Please include additional information regarding the survey or questionnaire used in the study and ensure that you have provided sufficient details that others could replicate the analyses. For instance, if you developed a questionnaire as part of this study and it is not under a copyright more restrictive than CC-BY, please include a copy, in both the original language and English, as Supporting Information.

Furthermore, please provide additional information regarding participant recruitment for the study, in particular please provide details regarding how potential participants were approached for the study and any eligibility criteria's applied.

When reporting the results of qualitative research, we suggest consulting the COREQ guidelines: http://intqhc.oxfordjournals.org/content/19/6/349. In this case, please consider including more information on the number of interviewers, their training and characteristics; and please provide the interview guide used. Additionally, please provide additional information regarding the interview guide development process, including the theories or frameworks which were employed.

5. Please provide additional details regarding participant consent. In the ethics statement in the Methods and online submission information, since consent was verbal/oral, please specify: 1) whether the ethics committee approved the verbal/oral consent procedure, 2) why written consent could not be obtained, and 3) how verbal/oral consent was recorded. If your study included minors, please state whether you obtained consent from parents or guardians in these cases. If the need for consent was waived by the ethics committee, please include this information.

Additional Editor Comments (if provided):

Reviewers' comments:

Reviewer's Responses to Questions

**Comments to the Author**

1. Is the manuscript technically sound, and do the data support the conclusions?

Reviewer #1: Yes

Reviewer #2: Yes

2. Has the statistical analysis been performed appropriately and rigorously? 

Reviewer #1: I Don't Know

Reviewer #2: Yes

3. Have the authors made all data underlying the findings in their manuscript fully available?

Reviewer #1: Yes

Reviewer #2: No

4. Is the manuscript presented in an intelligible fashion and written in standard English?

Reviewer #1: Yes

Reviewer #2: Yes

5. Review Comments to the Author

Reviewer #1: Very nice paper. Though I am not totally convinced that the SEIPS approach generates much new information that is not already explored in the WHO multimodal strategy and its associated tools, I do think that reworking the information around this model is always helpful for mapping exactly where the barriers and facilitators to each element (environment, people, etc.) lie, and shifts the description of the issue. I personally think that the way that the data was presented was especially helpful as it is easy to identify gaps where barriers do not match up with the facilitators (or where they do).

Minor Comments:

were the HH observers validated ? I would like some more information about what you mean that the observers were “unidentified”, was this to reduce the Hawthorne effect? Were they trying to remain hidden, or did they say that they were observing something else?

p 8-9

Interesting how initial perception of compliance is so much higher, and very interesting that there was no difference in scores of trained and untrained participant at baseline- can you provide information on what the training consisted of before your intervention?

p11 line 192- for the facilitators for “person”, do you mean “previous knowledge of HH”?

p15 line 254- Didi you mean 10% instead of 20%? (I thought baseline was 9.5%,but may have missed something)

p18 line 309- I would argue that glove use tends to lower hand hygiene, so this should be mentioned. Glove availability should be accompanied by training on appropriate glove use, and how to properly perform HH in scenarios where gloves are needed.

I recommend accepting this paper with a few minor revisions.

Reviewer #2: In abstract you need to clearly state what the aim/scope of the study is

In abstract identify the study timeline

At the end of the introduction/nackround you need to identify the study aim/scope clearly/ any hypothesis if any

The article is missing the number of total population of nurses and doctors, lab techs number of people working This should be made clear

Selection process/enrolment of study participants and sample size selection has not been clearly identified

Can the authors explain the education the observers received for HH training, was there any validation between observers conducted such as inter rater reliability between observers? what was the interval of observations?

The authors need to further research what has already been published in the hand hygiene literature to make better comparisons to the literature and to better place the impact of this study in the context of the body of literature already published

There is literature about hand hygiene and nurses/doctors Please refer again to discussion in Lines 266-275

A comparison to the literature would be helpful.

You state that dispensers were installed inside patient rooms, (line 127). Does this include hanging on the bed? Please be more specific.

It is not clear what time of day the observations were done and what this could imply.

You associate the post intervention improvement to the COVID-19 pandemic, however other than the influence of an it needs to be mentioned that there was lack of resources, lack of observations etc and no routine monitoring was conducted before this study at the hospital and how can this change be sustained after the pandemic

Please give 95% confidence intervals for outcome measures /proportions

6. PLOS authors have the option to publish the peer review history of their article (what does this mean?). If published, this will include your full peer review and any attached files.

Reviewer #1: **Yes: **Alexandra Peters

Reviewer #2: No

---

## [Author Response · Author response to Decision Letter 0]

28 Sep 2021

Response to Reviewer #1 comments:

Comment: Very nice paper. Though I am not totally convinced that the SEIPS approach generates much new information that is not already explored in the WHO multimodal strategy and its associated tools, I do think that reworking the information around this model is always helpful for mapping exactly where the barriers and facilitators to each element (environment, people, etc.) lie, and shifts the description of the issue. I personally think that the way that the data was presented was especially helpful as it is easy to identify gaps where barriers do not match up with the facilitators (or where they do).

Response: Authors express gratitude to the reviewer for their insight. 

Comment: Were the HH observers validated? I would like some more information about what you mean that the observers were “unidentified”, was this to reduce the Hawthorne effect? Were they trying to remain hidden, or did they say that they were observing something else?

Response: While the HH observers were not validated, they did receive HH training, which we have added to the manuscript in lines 140-141. We also removed the word “unidentified” to clarify that observers did not identify themselves as HH observers while conducting observations in order to reduce the Hawthorne effect (lines 146-148).

Comment: p8-9. Interesting how initial perception of compliance is so much higher, and very interesting that there was no difference in scores of trained and untrained participant at baseline- can you provide information on what the training consisted of before your intervention?

Response: We appreciate the reviewer’s thoughtful comment. JUMC did not provide systematic HH training prior to this study, and we did not explicitly ask questionnaire participants to write where they received prior HH training (i.e at another hospital, in school, etc). We did add more information about the lack of prior training offered at JUMC in lines 214-215.

Comment: p11 line 192- for the facilitators for “person”, do you mean “previous knowledge of HH”?

Response: We have edited line in the table to read “previous knowledge of HH”

Comment: p15 line 254- Didi you mean 10% instead of 20%? (I thought baseline was 9.5%, but may have missed something)

Response: We have edited line 324 to read “10%.”

Comment: p18 line 309- I would argue that glove use tends to lower hand hygiene, so this should be mentioned. Glove availability should be accompanied by training on appropriate glove use, and how to properly perform HH in scenarios where gloves are needed.

Response: We agree with this addition. Please see lines 414-416 for discussion of the necessity of training on appropriate glove use. 

Response to Reviewer #2 comments:

Comment: 

In abstract you need to clearly state what the aim/scope of the study is

In abstract identify the study timeline

Response: We have edited our abstract to read “We aimed to apply the Systems Engineering Initiative for Patient Safety (SEIPS) model to increase effectiveness and sustainability of the World Health Organization’s (WHOs) hand hygiene (HH) guidelines within healthcare systems” (line 27-29). Additionally, in line 31, we added the study timeline to our abstract. 

Comment: At the end of the introduction/nackround you need to identify the study aim/scope clearly/ any hypothesis if any

Response: We edited our Background section (lines 72-90) to read, “We aimed to apply the SEIPS model to increase effectiveness and sustainability of HH interventions at Jimma University Medical Center (JUMC) in Jimma, Ethiopia.”

Comment: The article is missing the number of total population of nurses and doctors, lab techs number of people working This should be made clear

 Response: We added the total population of nurses, doctors, and lab techs working at JUMC in lines 100-101. 

Comment: Selection process/enrolment of study participants and sample size selection has not been clearly identified

 Response: We added more information on selection process for questionnaire and interview participants in lines 126-127 and 150-152 respectively. We also described how sample size was determined for questionnaires and interviews in lines 127-134 and 152-153 respectively (and added a reference to further highlight the sample size needed for qualitative interviews). 

Comment: Can the authors explain the education the observers received for HH training, was there any validation between observers conducted such as inter rater reliability between observers? what was the interval of observations?

 Response: While there we did not do any validation between observers, observers did complete HH training, which we added to the manuscript in lines 140-141. We have also edited line 141 to describe the interval of observations. 

Comment: The authors need to further research what has already been published in the hand hygiene literature to make better comparisons to the literature and to better place the impact of this study in the context of the body of literature already published

 Response: We thank the reviewer for their suggestion. We have edited our Discussion section to more explicitly compare our study results to results from previous studies on HH in Ethiopia (see lines 335-353). We feel we have conducted a thorough review of studies conducted on HH in Ethiopia and made efforts to interact with the literature in our Discussion section (lines 324-361).

Comment: There is literature about hand hygiene and nurses/doctors Please refer again to discussion in Lines 266-275

A comparison to the literature would be helpful.

Response: We compare our finding that baseline HH compliance was higher in physicians than nurses to three other studies from Ethiopia (335-51). We added one reference to a study that discusses potential reasons for higher observed compliance amongst physicians than nurses in other LMICs (lines 351-353). We feel we adequately compare our compliance data to existing literature on HH in Ethiopia specifically. 

Comment: You state that dispensers were installed inside patient rooms, (line 127). Does this include hanging on the bed? Please be more specific.

Response: We revised line 181 to clarify the IPC team installed 400 hand rub dispensers on the walls inside of patient rooms and in ward hallways, not hanging on patient beds. 

Comment: It is not clear what time of day the observations were done and what this could imply.

Response: We edited line 141 to highlight that HH observations were conducted during the daytime Monday-Friday. While HH compliance may admittedly be lower overnight than during the daytime due to HCW fatigue, personnel limitations restricted our ability to conducted observations 24/7. Methods for observations were consistent between the baseline and follow-up periods, ensuring that the time of day of the observations did not contribute to the overall increase in HH compliance between the follow-up and baseline observations.

Comment: You associate the post intervention improvement to the COVID-19 pandemic, however other than the influence of an it needs to be mentioned that there was lack of resources, lack of observations etc and no routine monitoring was conducted before this study at the hospital and how can this change be sustained after the pandemic

Response: We appreciate the suggestion to mention the need for sustaining improvements in HH after Covid-19. Please see lines 362-364, where we noted that additional interventions are needed to sustain improvements in HH resources, monitoring, and training after Covid.

Comment: Please give 95% confidence intervals for outcome measures /proportions

Response: To provide more statistical analysis of our observation data, we calculated the odds ratio for baseline and follow-up total HH compliance and have reported this value with a 95% confidence interval (line 294-295). We do not feel any further statistical analysis of our observation data would contribute to the overall goal and scope of our paper.

---

## [Editor Report · Decision Letter 1]

4 Oct 2021

Utilizing the SEIPS model to guide hand hygiene interventions at a tertiary hospital in Ethiopia

PONE-D-21-07299R1

Dear Ms. Berman,

We’re pleased to inform you that your manuscript has been judged scientifically suitable for publication and will be formally accepted for publication once it meets all outstanding technical requirements.

Kind regards,

Monika Pogorzelska-Maziarz

Academic Editor

PLOS ONE

---

## [Editor Report · Acceptance letter]

19 Oct 2021

PONE-D-21-07299R1 

Utilizing the SEIPS model to guide hand hygiene interventions at a tertiary hospital in Ethiopia 

Dear Dr. Berman:

I'm pleased to inform you that your manuscript has been deemed suitable for publication in PLOS ONE. Congratulations! Your manuscript is now with our production department. 

Kind regards, 

on behalf of

Dr. Monika Pogorzelska-Maziarz 

Academic Editor

PLOS ONE